# Large-Cell Neuroendocrine Carcinoma of the Cervix: Case Report and Literature Review

**DOI:** 10.3390/diagnostics15060775

**Published:** 2025-03-19

**Authors:** Wing Yu Sharon Siu, Chiu-Hsuan Cheng, Dah-Ching Ding

**Affiliations:** 1Department of Obstetrics and Gynecology, Hualien Tzu Chi Hospital, Buddhist Tzu Chi Medical Foundation, Tzu Chi University, Hualien 970, Taiwan; wingyusharonsiu@gmail.com; 2Department of Pathology, Hualien Tzu Chi Hospital, Buddhist Tzu Chi Medical Foundation, Tzu Chi University, Hualien 970, Taiwan; chiuhsuan.cheng@gmail.com; 3Institute of Medical Sciences, Tzu Chi University, Hualien 970, Taiwan

**Keywords:** large cell, neuroendocrine carcinoma, cervix, multimodal, prognosis

## Abstract

**Background and clinical significance**: Large-cell neuroendocrine carcinoma (LCNEC) of the cervix is considered a rare type of cancer: it represents <1% of invasive cervical cancers. The optimal treatment protocol is not fully established because of its rarity and diagnostic challenges. **Case Presentation**: A 72-year-old Asian female presented to our outpatient clinic with postmenopausal vaginal spotting for 1 month. Vaginal sonography revealed a cervical tumor of 2.7 cm in diameter with hypervascularity. Tumor markers such as CA 125, CA 19-9, carcinoembryonic antigen, and squamous cell carcinoma antigen all showed no abnormality. Due to high suspicion of cervical cancer, a pap smear and endocervical curettage were performed and confirmed the diagnosis of LCNEC. A positron emission tomography–computed tomography scan demonstrated a glucose hypermetabolic lesion in the mid-pelvic region, localized to the uterus, consistent with LCNEC. Surgery with radical hysterectomy, bilateral salpingo-oophorectomy, and bilateral pelvic lymph node dissection was performed. The patient was finally diagnosed with pT1b2N1mi, FIGO IIIC1. Immunohistochemical stain shows that the neoplastic cells were CK (+), p63 (−), p16 (−), CEA (−), vimentin (−), ER (−), WT-1 (−), p53 (−), and CD56 (+), with a high Ki67 index (75%). Concurrent chemotherapy with cisplatin and radiotherapy was performed. Four cycles of etoposide and cisplatin were planned. A 3-month follow-up of this patient revealed stable tumor marker levels. **Conclusions**: This case highlights the diagnostic challenges and aggressive nature of LCNEC of the cervix, emphasizing the need for a standardized treatment approach to improve patient outcomes.

## 1. Introduction

Large-cell neuroendocrine carcinoma (LCNEC) of the cervix is a rare and aggressive malignancy, accounting for approximately 0.6% of invasive cervical cancers [1]. Patients are typically young, with a median age of 37–41 years [2,3].

Despite often presenting at early stages, LCNEC has a poor prognosis, with median overall survival ranges of 16.5–26 months and 5-year survival rates of 29–36% [2,3,4]. Early-stage disease and a lower FIGO stage are associated with improved survival [2,3].

Histologically, LCNEC of the cervix is characterized by large tumor cells arranged in an organoid pattern, with immunoreactivity for neuroendocrine markers [1]. Diagnosis can be challenging due to its rarity and similarity to other cervical cancers [5]. Cytological features include ball-like tumor cell clusters, rosettoid patterns, and nuclear molding [6]. Histologically, LCNECs may present as mixed adenoneuroendocrine carcinomas with solid and glandular areas [7]. Immunohistochemistry is crucial to diagnosis, with tumors typically expressing neuroendocrine markers like CD56, synaptophysin, and chromogranin [5].

Treatment approaches vary; however, surgery, particularly with lymphadenectomy, significantly improves survival [3]. Chemotherapy, especially platinum-based regimens, may also enhance outcomes [2]. However, the rarity of LCNEC makes establishing optimal treatment protocols challenging [8]. Given its aggressive nature, multimodal therapy should be considered for LCNEC patients [8].

We aimed to report a case with LCNEC of the cervix treated with radical hysterectomy with bilateral salpingo-oophorectomy and bilateral pelvic and para-aortic lymph node dissection, concurrent chemoradiation, and chemotherapy.

## 2. Case Presentation

A 72-year-old Asian female without underlying diseases presented with postmenopausal vaginal spotting for 1 month. Initial evaluation at a local hospital detected a 3 cm cervical tumor, leading to referral for further investigation. Endocervical curettage confirmed LCNEC of the cervix. She reported no other symptoms, and her medical history was unremarkable except for menopause at 53 years and three vaginal deliveries. Physical examination revealed vaginal staining with a small tumor dropped from the endocervix, and laboratory tests showed normal blood counts, liver, and kidney function, with tumor markers (CA 125, CA19-9, CEA, and SCC) within normal limits. Imaging studies, including vaginal ultrasound (Figure 1A,B) and positron emission tomography–computed tomography (PET-CT) (Figure 1C,D), identified a hypermetabolic 2.7 cm cervical tumor confined to the uterus. A cervical biopsy was performed, and LCNEC of the cervix was diagnosed.

She underwent open type C2 radical hysterectomy (Querleu–Morrow classification), bilateral salpingo-oophorectomy and pelvic and para-aortic lymph node dissection (Figure 2). Histopathology confirmed a poorly differentiated (Grade 3) LCNEC with lymphovascular invasion and right pelvic lymph node metastasis.

Hematoxylin and eosin staining show a nested and trabecular growth pattern with small, hyperchromatic tumor cells, scant cytoplasm, and frequent mitotic activity, suggesting an aggressive malignancy (Figure 3A,B). Synaptophysin (Figure 3C) and CD56 (Figure 3D) immunohistochemistry demonstrate strong positive staining, confirming neuroendocrine differentiation. The presence of necrosis and high cellularity further supports the diagnosis of a poorly differentiated NEC of the cervix. The final diagnosis was LCNEC of the cervix, pT1b2N1mi, FIGO stage IIIC1.

Postoperatively, she received concurrent chemotherapy and radiotherapy, including weekly cisplatin and external beam radiotherapy (4500 cGy) plus intravaginal radiotherapy (2100 cGy). She experienced mild fatigue but tolerated the treatment well.

Planned adjuvant chemotherapy included four cycles of etoposide and cisplatin. Four months post-surgery, the patient remained in remission with no signs of recurrence.

## 3. Literature Review and Discussion

### 3.1. Search Strategy

This systematic review followed the Preferred Reporting Items for Systematic Reviews and Meta-Analyses (PRISMA) 2020 guidelines [9].

A systematic search used “large cell, neuroendocrine carcinoma, uterine cervix” from inception to 10 February 2025. Synonyms and related terms were also included to expand the scope. The bibliographies of relevant reviews and included studies were also examined. Table 1 provides an overview of the search strategy used for the PubMed, Scopus, Web of Science, and Embase databases.

Initially, 877 articles were extracted from the databases, 563 of which were removed because of duplication. The remaining 314 articles were reviewed based on their titles and abstracts, and 214 were removed because of irrelevance. Subsequently, 100 articles were reviewed based on our exclusion criteria. Finally, 100 articles met the inclusion criteria and were included in the systematic review (Figure 4).

### 3.2. Epidemiology and Risk Factors

#### 3.2.1. Prevalence and Incidence of LCNEC of the Cervix

LCNEC of the uterine cervix is a rare and aggressive malignancy, accounting for <5% of all cervical cancers [10]. In a study of 972 invasive cervical carcinoma cases, only 6 (0.6%) were identified as LCNECs [1]. Another study found 14 (3.5%) neuroendocrine carcinomas among 389 primary cervical carcinomas, with 3 cases classified as LCNECs [11].

#### 3.2.2. Association with Human Papillomavirus (HPV) and Other Risk Factors

High-risk HPV, particularly types 16 and 18, is frequently associated with LCNEC [12]. In some cases, LCNEC may coexist with cervical intraepithelial neoplasia (CIN), both lesions potentially harboring HPV 16 DNA [13]. However, some cases may be HPV-negative [14]. LCNEC can coexist with other cervical cancer types, such as squamous cell carcinoma [14]. Risk factors for lymph node metastasis in cervical cancer include preoperative anemia, deep stromal invasion, absent or slight inflammatory reaction, and keratinizing squamous cell carcinoma [15].

### 3.3. Diagnosis and Histopathology

#### 3.3.1. Clinical Presentation

LCNEC of the cervix typically presents with vaginal bleeding or abnormal pap smears, with a median age of 36 years at diagnosis [1,16,17,18]. Patients typically present with vaginal bleeding and pelvic pain [19,20,21]. The cytological examination may reveal large, loosely cohesive cells with nuclei 3–5 times larger than small lymphocytes [18,22,23]. The disease often presents at an advanced stage with metastases to lymph nodes, lungs, liver, and bones [20,24]. One study reported that 75% of reported cases (9/12) are stage Ib [18]. LCNEC of the uterine cervix is a rare and aggressive malignancy with a poor prognosis even in the early stage [15,25,26].

#### 3.3.2. Imaging and Diagnostic Modalities

Magnetic resonance imaging (MRI) demonstrated higher sensitivity than PET/CT for detecting metastatic lymph nodes in cervical cancer patients [27]. PET/MRI showed superior diagnostic accuracy (94.90%) for cervical cancer staging compared with PET/CT, MRI, and CT [28]. It also exhibited higher detection rates for various types of invasion and greater sensitivity, specificity, and accuracy in diagnosing lymph node metastasis [28]. MRI is the preferred method for local cervical cancer evaluation, whereas CT is effective for assessing extrauterine spread [29]. PET/CT demonstrates high diagnostic performance in detecting tumor relapse and metastatic lymph nodes [29]. These imaging modalities play crucial roles in accurate staging, which is essential to the optimal management and prognosis of cervical cancer patients [29,30].

Colposcopy plays a crucial role in evaluating cervical lesions and detecting precancerous and cancerous conditions. Studies have shown that colposcopy with biopsy reduces unnecessary surgical procedures and decreases positive margin rates in large loop excision of the transformation zone (LLETZ) [31]. Colposcopy demonstrates high sensitivity (93.33–98.30%) in detecting CIN, although specificity varies (57.30–89.74%) [32,33]. Cytology (pap smear) complements colposcopy, with reported sensitivity ranging from 15 to 50% and specificity from 89.74 to 98.4% [33]. For rare conditions like LCNEC of the cervix, cytological and colposcopic findings are valuable for early diagnosis, as these tumors have poor prognoses [22]. Combining colposcopy, biopsy, and cytology provides a comprehensive approach to evaluating cervical abnormalities and guiding appropriate management.

#### 3.3.3. Histopathology and Immunohistochemistry

Cytologically, LCNECs exhibit large cells with coarse chromatin, prominent nucleoli, and frequent mitotic figures [34]. Histologically, they show trabecular and organoid growth patterns with extensive necrosis [19]. LCNEC is characterized by large cells with prominent nucleoli, high mitotic activity, and neuroendocrine differentiation [35]. LCNEC is characterized by trabecular and organoid growth patterns, extensive necrosis, and positive immunoreactivity for neuroendocrine markers like synaptophysin [18,19]. LCNEC is characterized by large tumor cells arranged in an organoid growth pattern, with immunoreactivity for neuroendocrine markers such as chromogranin A and synaptophysin [1,11,18]. Some LCNECs may exhibit TTF1 immunoreactivity, which is important for accurate diagnosis and appropriate treatment [35]. Immunohistochemically, LCNECs are positive for neuroendocrine markers such as synaptophysin and CD56 [34]. The diagnosis of LCNEC relies on a combination of clinical, cytological, histological, and immunohistochemical findings, with neuroendocrine marker positivity being crucial to confirmation [19,34].

LCNEC can occur in combination with small-cell carcinoma or other cervical malignancies [11,34]. Differential diagnosis includes small-cell neuroendocrine carcinoma and other cervical tumors, such as adenocarcinomas, sarcomas, and metastatic carcinomas [36].

NECs of the uterine cervix are rare and aggressive, comprising small-cell (SCNEC) and LCNEC subtypes [11,23]. Diagnosis requires histopathology and immunohistochemistry, with NECs typically expressing chromogranin, synaptophysin, NSE, and CD56 [11]. Morphologically, SCNECs have smaller nuclear diameters compared with LCNECs [37]. Genetic differences between SCNEC and LCNEC have been observed, including variations in allelic losses and gene expression patterns, suggesting they may be distinct entities despite some overlapping features [37]. The frequencies of the expression of CD56, mASH1, TTF-1, and p16 were higher and that of NeuroD was lower in SCNEC than in LCNEC [37]. Allelic losses at D5S422 (5q33) were more frequent in SCNEC than in LCNEC [37].

### 3.4. Treatment Strategies

#### 3.4.1. Surgery

Treatment options include surgery, radiotherapy, and chemotherapy, but prognosis remains poor, with patients often succumbing to the disease within 6–18 months of diagnosis [19,20]. Radical hysterectomy with pelvic lymphadenectomy (RHPL) has been the standard treatment for early-stage cervical cancer, including neuroendocrine carcinomas [18,38,39]. However, some studies suggest that a less radical approach may be sufficient for certain patients [38]. RHPL has shown favorable long-term outcomes with minimal morbidity, although factors like advanced stage, nonsquamous histology, and nodal involvement are associated with poorer prognosis [39]. Laparoscopic RHPL has emerged as a safe alternative with reduced surgical morbidity and shorter hospital stays [40]. However, advanced-stage neuroendocrine carcinomas have poor outcomes despite various treatment approaches [23].

The LACC trial demonstrated that minimally invasive surgery (MIS) for early-stage cervical cancer is associated with higher recurrence and mortality rates compared with open surgery [41,42]. This finding has led to a reevaluation of surgical approaches in cervical cancer management, with open radical hysterectomy now recommended as the standard of care for stage IA2-IB1 cervical cancer [41]. However, the adoption of this recommendation varies across Asian countries, with some still performing MIS in a significant proportion of cases [43]. Notably, MIS without uterine manipulator or with vaginal cuff closure showed similar recurrence rates to open surgery, suggesting potential modifications to improve MIS outcomes [43]. The LACC trial results have prompted further research and discussions among gynecologic oncologists worldwide, as evidenced by the Korean Society of Gynecologic Oncology survey [44].

#### 3.4.2. Chemotherapy

Adjuvant therapy, including etoposide–platinum or irinotecan–platinum regimens, has demonstrated higher response rates than taxane–platinum regimens [45]. Locally advanced disease, para-aortic node metastasis, distant metastasis, and insufficient chemotherapy cycles are associated with poor survival [45]. Despite the generally poor prognosis, some patients with early-stage disease treated with surgery and adjuvant chemotherapy have shown long-term survival [23,46].

Neoadjuvant chemotherapy (NAC) followed by RH shows promise in treating LCNEC of the uterine cervix. One case study reported successful treatment using irinotecan plus cisplatin as NAC [47]. Another study found that postoperative chemotherapy with irinotecan and cisplatin led to complete remission in a patient with LCNEC [23]. A multicenter retrospective study demonstrated improved overall survival with NAC followed by RH compared with RH alone in locally advanced nonsquamous cervical carcinomas, particularly for mucinous adenocarcinomas [48]. For early-stage NEC, multimodality therapy, including NAC, surgery, and adjuvant therapy, showed potential for long-term survival, especially in patients with lymph node metastasis and large tumors [49].

#### 3.4.3. Radiotherapy

For locally advanced NEC, brachytherapy combined with external beam radiation therapy (EBRT) significantly improves overall survival compared with EBRT alone [50]. No difference in overall survival was noted between patients treated with NAC and those who received concurrent chemoradiation [50]. Chemoradiation may be superior to surgery for early-stage, node-negative disease [51].

Due to its rarity, optimal treatment protocols are not well established, but multimodal therapy is recommended [8].

#### 3.4.4. Targeted Therapy and Immunotherapy

Recent studies have explored the potential of PARP inhibitors for treating LCNEC based on next-generation sequencing (NGS) results. In two case reports, patients with advanced LCNEC harboring BRCA mutations were treated with PARP inhibitors, resulting in disease stabilization and prolonged survival [52,53]. One patient achieved 74 months of disease stabilization with rucaparib treatment [53]. These findings suggest that NGS-guided PARP inhibitor therapy may offer a promising treatment option for some LCNEC patients, particularly those with BRCA mutations, potentially improving outcomes in this challenging disease.

Immune checkpoint inhibitors (ICIs) have shown promising results in treating LCNEC and neuroendocrine carcinoma of the cervix (NECC). Case reports demonstrate complete responses to nivolumab in PD-L1-negative SCNEC of the cervix [54] and durable responses to pembrolizumab in LCNEC with a high tumor mutation burden [55]. These findings suggest that ICIs may be effective even in PD-L1-negative tumors. While the evidence is limited to individual case reports and small series, ICIs show potential for dramatic responses in a subset of patients [56]. PD-1/PD-L1 inhibitors are being explored as monotherapy and in combination with other treatments for NECC, offering a new direction for immune-targeted therapy [57]. Ipilimumab–nivolumab combination immunotherapy showed a durable response in patients with recurrent neuroendocrine carcinoma of the cervix [58]. However, further studies are needed to confirm the efficacy of ICIs and establish their role as a standard treatment strategy in LCNEC and NECC [59].

#### 3.4.5. Clinical Trials

We searched by using “large cell neuroendocrine carcinoma of cervix” as keywords to identify relevant clinical trials listed on clinicaltrials.gov. A total of three trials were conducted up to 18 February 2025. Of these, one study is currently open, while the remaining are completed, closed, or terminated.

This study evaluates the efficacy of bevacizumab and paclitaxel in patients with recurrent small-cell, large-cell, and NEC cervical and uterine cancers, focusing on progression-free survival, overall survival, response rates, quality of life, and treatment toxicity [NCT00626561]. Additionally, it aims to correlate clinical data with patient outcomes to improve the understanding of neuroendocrine cervical cancer [NCT04723095]. Another trial investigates INCAGN02385 to assess its safety, tolerability, and preliminary efficacy in advanced malignancies [NCT03538028].

### 3.5. Prognosis and Outcomes

#### 3.5.1. Survival Rates

Patients typically present at a young median age of 37–41 years, often with early-stage disease [2,3]. Despite this, LCNEC has a high recurrence rate and frequently metastasizes [8,51]. Interestingly, LCNEC may have better outcomes compared with other neuroendocrine subtypes [51]. For all patients with NEC of the cervix, the 5-year event-free survival (EFS) rate was 20%, and the 5-year overall survival (OS) rate was 27% [51]. Patients with LCNEC of the cervix had a significantly longer median EFS (median not reached vs. 10.0 months) and a trend toward improved OS (153 months vs. 21 months) compared with those with other histologic types [51].

The prognosis is poor, with a median overall survival of 16 months for cervical LCNEC [16]. The prognosis is generally poor, with mean disease-free intervals of 17.5 months reported [11]. However, polypoid NECs or those arising from polyps may have a more favorable prognosis [60]. Five-year OS in patients with classic large-cell carcinoma and LCNEC in stage I was 67 and 73%, respectively [37]. Patients with NeuroD expression had better survivals, and those with p63 expression had poorer survivals in LCNEC [37].

#### 3.5.2. Prognostic Factors: Stage at Diagnosis, Treatment Modality, Lymph Node Involvement, and Ki-67 Index

The overall median survival was 16.5 months, with survival decreasing as the stage advanced [16]. Multivariate analysis revealed that at an earlier stage of diagnosis, the addition of chemotherapy was a significant prognostic factor associated with improved survival [2]. Specifically, platinum-based chemotherapy, either alone or combined with etoposide, demonstrated a survival advantage [2]. The study concluded that perioperative chemotherapy, particularly platinum-based regimens with or without etoposide, could improve survival outcomes in patients with LCNEC [2]. These findings highlight the importance of early diagnosis and the potential benefit of incorporating platinum-based chemotherapy into treatment strategies for LCNEC [2].

LCNEC often presents with lymph node involvement and distant metastasis, leading to poor prognosis [8,25]. Lymph node metastasis and FIGO stage are independent prognostic factors [61]. The aggressive nature of LCNEC is evident even in early-stage disease, with frequent recurrences and distant metastases [8].

Immunohistochemistry typically shows strong positivity for neuroendocrine markers and a high Ki67 proliferation index [25]. However, the link between Ki67 and prognosis is not known.

#### 3.5.3. Recurrence Patterns and Metastasis Sites

Patients with LCNEC tend to have better outcomes than other NECC subtypes [51]. NECC frequently recurs within 3 years of initial treatment, with distant metastases being common [62]. Recurrence patterns vary, with reported metastases to the lung, breast, and retroperitoneum [63]. The number of recurrent sites and abdominal organ recurrence are independent prognostic factors for postrecurrence survival [62].

### 3.6. Current Limitations on Diagnosis and Treatment

#### 3.6.1. Diagnostic Limitation

Due to its rarity, many clinicians and pathologists have limited experience, leading to frequent misdiagnosis. Histopathological confirmation requires neuroendocrine differentiation, but routine H&E staining may not be sufficient, necessitating immunohistochemical markers such as synaptophysin, chromogranin A, and CD56 [18]. However, inconsistent marker expression and a lack of standardized diagnostic criteria further complicate accurate identification. Molecular profiling, including TP53 and RB1 mutations, has shown a potential to improve diagnostic accuracy, but its clinical utility remains unstandardized [64]. Additionally, the absence of a universally accepted grading system makes it challenging to differentiate LCNEC from small-cell neuroendocrine carcinoma and other poorly differentiated tumors [5].

#### 3.6.2. Therapeutic Limitations

Therapeutic options for LCNEC are limited, as there are no standardized treatment guidelines, and current approaches are often extrapolated from small-cell neuroendocrine carcinoma or pulmonary neuroendocrine tumors [65]. Radical hysterectomy is commonly performed, but surgery alone is insufficient due to early systemic dissemination [66]. Platinum-based chemotherapy (cisplatin/carboplatin with etoposide or irinotecan) remains the mainstay of treatment but is often ineffective in achieving long-term remission [59]. The role of adjuvant radiation therapy is unclear, with inconsistent survival benefits [59]. Emerging treatments, including immune checkpoint inhibitors and targeted therapies, have shown promise, but clinical data are lacking [57]. Given the high recurrence rate and poor prognosis, future research should focus on refining molecular diagnostics, conducting prospective clinical trials, and exploring novel therapeutic strategies to improve patient outcomes.

### 3.7. Perspectives

Current management typically involves platinum-based chemotherapy, but outcomes remain dismal [67]. Recent advances in molecular profiling have revealed potential therapeutic targets and genetic subcategories of LCNEC, offering promise for personalized therapies [67,68]. Immunotherapy agents, such as immune checkpoint inhibitors, have shown efficacy in related cancers and may hold potential for LCNEC treatment [69]. Future research directions include further genomic and molecular characterization of gynecological LCNEC to elucidate oncogenic pathways and driver mutations [16]. Collaborative efforts and establishing LCNEC-specific biobanks are essential to advancing our understanding of disease biology and developing targeted therapies [69].

## 4. Conclusions

LCNEC is a rare and highly aggressive malignancy with a poor prognosis despite multimodal treatment approaches. Diagnosis relies on histopathological and immunohistochemical findings, often showing high Ki-67 proliferation index and neuroendocrine marker expression. While platinum-based chemotherapy remains the mainstay of treatment, surgical resection, radiotherapy, and emerging targeted therapies, including immune checkpoint inhibitors and PARP inhibitors, offer potential avenues for improving patient outcomes. However, due to the high recurrence rate and limited survival even with aggressive treatment, there is a pressing need for further research into molecular and genomic profiling to identify new therapeutic targets. Future studies should focus on refining treatment strategies, enhancing early detection, and expanding clinical trials to improve survival rates and quality of life for patients with LCNEC.

## Figures and Tables

**Figure 1 diagnostics-15-00775-f001:**
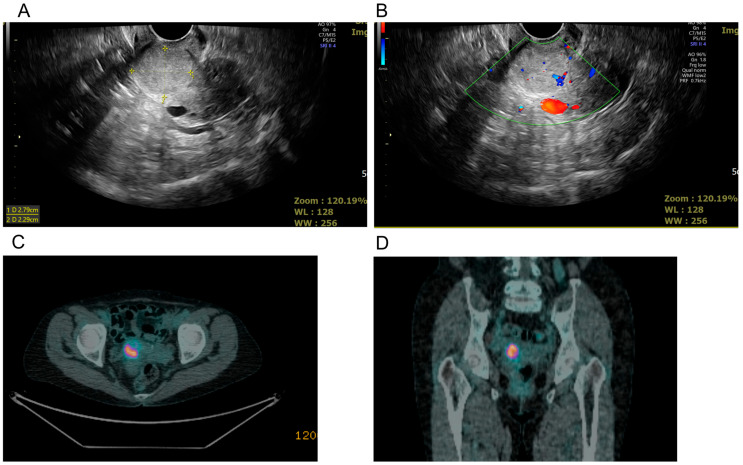
Multimodal imaging of pelvic lesion. (**A**) Transvaginal grayscale ultrasound image showing a well-defined solid mass measuring approximately 2.79 × 2.29 cm in the pelvic region (+..+1: dimension 1). (**B**) Transvaginal color Doppler ultrasound demonstrated increased vascularization within the lesion, suggesting neovascularity. (**C**) Axial PET/CT fusion image highlights hypermetabolic activity within the lesion, indicating potential malignancy. (**D**) Coronal PET/CT fusion image further localizes the hypermetabolic pelvic mass about surrounding anatomical structures.

**Figure 2 diagnostics-15-00775-f002:**
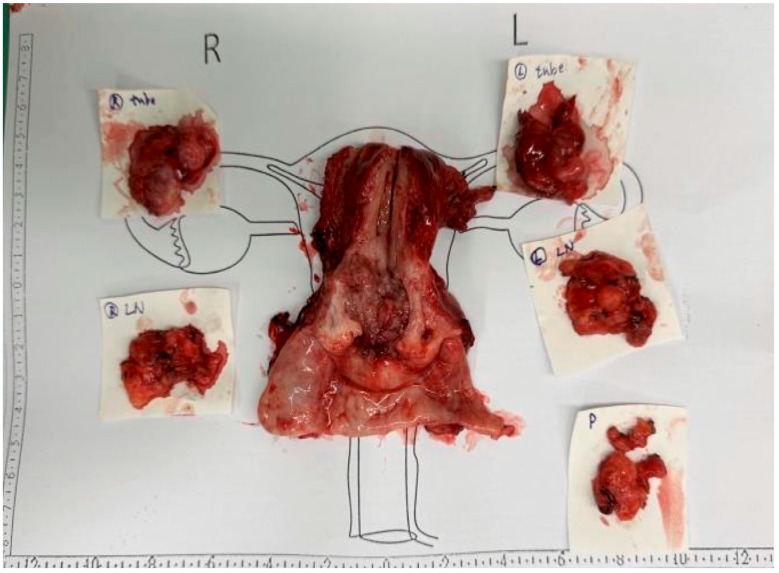
Gross pathology of the resected specimen. Macroscopic view of the surgically resected uterus, bilateral adnexa, and associated lymph nodes. The central specimen represents the uterus with an infiltrative tumor involving the endometrium, myometrium, and cervix. The surrounding excised tissues include the right (R) and left (L) adnexa and lymph nodes (LN), as well as an additional para-aortic (P) lymph node sample. The specimens are placed on a surgical mapping sheet for anatomical reference.

**Figure 3 diagnostics-15-00775-f003:**
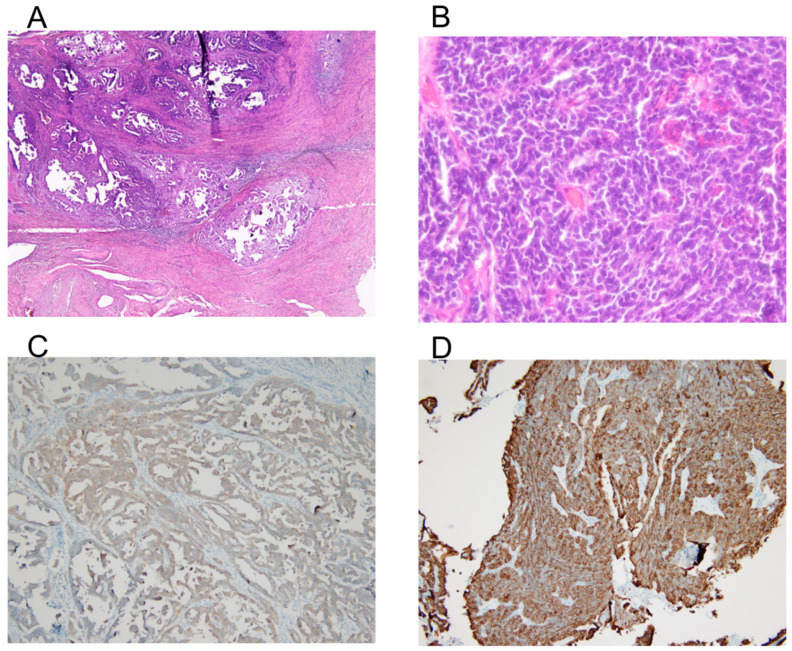
Histology and immunohistochemistry of the tumor. (**A**) Hematoxylin and eosin (H&E) staining at 40× magnification shows tumor architecture and cellular morphology. (**B**) High-power view (400× magnification) of the H&E-stained section, highlighting the tumor’s cellular features. A nested and trabecular growth pattern with small, hyperchromatic tumor cells, scant cytoplasm, and frequent mitotic activity suggests an aggressive malignancy. (**C**) Immunohistochemical staining for synaptophysin at 100× magnification, demonstrating positive expression in tumor cells, indicative of neuroendocrine differentiation. (**D**) Immunohistochemical staining for CD56 at 100× magnification, showing strong membranous expression, further supporting the neuroendocrine nature of the tumor.

**Figure 4 diagnostics-15-00775-f004:**
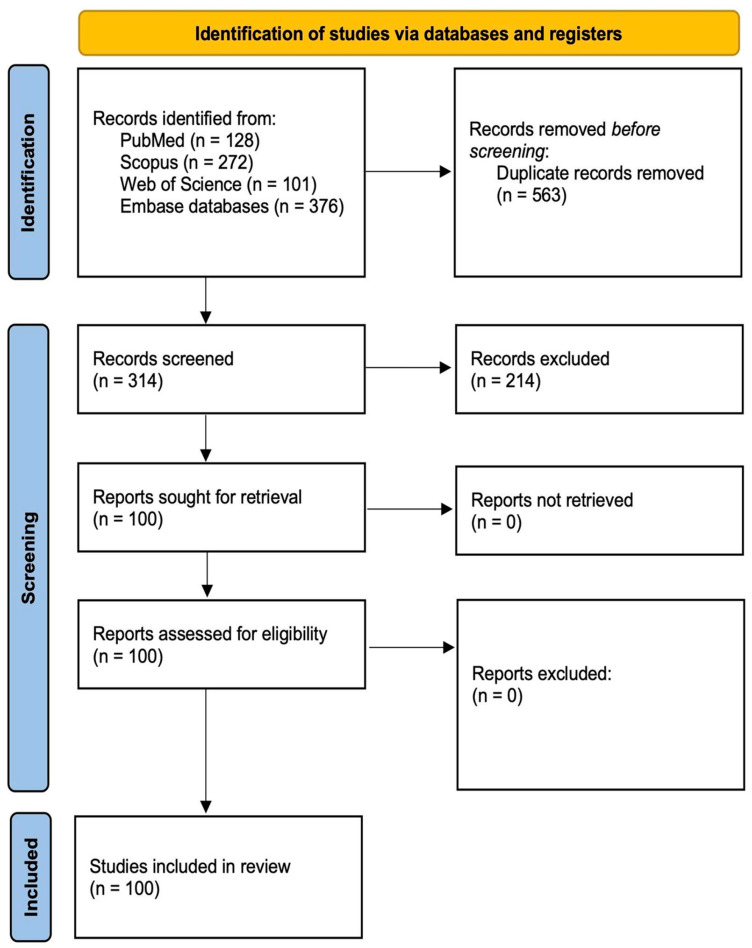
Study flowchart.

**Table 1 diagnostics-15-00775-t001:** Search strategy for the literature.

Items	Specifications
Timeframe	From inception to 10 February 2025
Database	PubMed, Scopus, Web of Science, and Embase
Search terms used	“Large cell, neuroendocrine carcinoma, uterine cervix”
Inclusion and exclusion criteria	All references were SCI-indexed articles written in English
Selection process	Two independent reviewers evaluated the titles and abstracts to determine eligibility

## Data Availability

All data are included in this article.

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
