# Peer review of "Large-Cell Neuroendocrine Carcinoma of the Cervix: Case Report and Literature Review"

_diagnostics, 2025, doi:10.3390/diagnostics15060775_

Round 1
Reviewer 1 Report
Comments and Suggestions for Authors -Authors work” Large cell neuroendocrine carcinoma of the cervix: a case report and literature review”-A well -written case report
-The discussion is good, but when I scan the literature. There is a good article “Large Cell Carcinoma of the Uterine Cervix: A Clinicopathologic Study of 12 Cases”, authors should add this to discussion part.
-P values in the article should be removed. I think it will be better.
After revision, final decision acceptable
SincerelyAuthor Response
-Authors work” Large cell neuroendocrine carcinoma of the cervix: a case report and literature review”
-A well -written case report
-The discussion is good, but when I scan the literature. There is a good article “Large Cell Carcinoma of the Uterine Cervix: A Clinicopathologic Study of 12 Cases”, authors should add this to discussion part.
Response: We thank the reviewer’s comment. We have added this article to the discussion part. (reference number 18, page 6, lines 148)
The statements read as:”One study reported that 75% of reported cases (9/12) are stage Ib [18].”
-P values in the article should be removed. I think it will be better.
Response: We thank the reviewer’s comment. We have removed p values in the discussion section.
After revision, final decision acceptable
Reviewer 2 Report
Comments and Suggestions for Authors
Large Cell Neuroendocrine Carcinoma of the Cervix: A Case Report and Literature Review
Overall Recommendation: Minor Revision
This manuscript presents a rare case report of a large cell neuroendocrine carcinoma (LCNEC) of the cervix, coupled with an extensive literature review. The case is well-documented with comprehensive clinical, imaging, and histopathological data, and the literature review covers epidemiology, diagnostic challenges, various treatment modalities, and prognostic factors associated with LCNEC. Overall, the manuscript addresses an important yet rare clinical entity and has the potential to add value to the existing literature on cervical neuroendocrine carcinomas.
I suggest to improve and clarify the description of the methodology for the literature search. When discussing the evidences, I suggest to intensify the critics. Since there is not a clear strategy in the review of the literature, there is no real discussion regarding the limitations of current treatment modalities and diagnostic approaches.
correct some typos, enhance figures legend.
specify what kind “radical hysterectomy” they did according to the Querleu-Morrow classification (e.g., Type B, C, or D) and discuss around the approach (MIS vs open) considering that the cervical tumor measured 2.7 cm in diameter with hypervascularity given the evidence from the LACC trial.
Author Response
This manuscript presents a rare case report of a large cell neuroendocrine carcinoma (LCNEC) of the cervix, coupled with an extensive literature review. The case is well-documented with comprehensive clinical, imaging, and histopathological data, and the literature review covers epidemiology, diagnostic challenges, various treatment modalities, and prognostic factors associated with LCNEC. Overall, the manuscript addresses an important yet rare clinical entity and has the potential to add value to the existing literature on cervical neuroendocrine carcinomas.
I suggest to improve and clarify the description of the methodology for the literature search.
Response: We thank the reviewer’s comment. We have added the methodology for the literature search (Figure 4). (section 3.1, page 4, lines 109-121)
The statements read as:”3.1. Search strategy
This systematic review followed the Preferred Reporting Items for Systematic Reviews and Meta-Analyses (PRISMA) 2020 guideline [9].
A systematic search used “large cell, neuroendocrine carcinoma, uterine cervix” from inception to 10 February 2025. Synonyms and related terms were also included to expand the scope. The bibliographies of relevant reviews and included studies were also examined. Table 1 provides an overview of the search strategy used for the PubMed, Scopus, Web of Science, and Embase databases.
Initially, 877 articles were extracted from the databases, of which 563 were removed because of duplication. The remaining 314 articles were reviewed based on their titles and abstracts, and 214 were removed because of irrelevance. Subsequently, 100 articles were reviewed based on our exclusion criteria. Finally, 100 articles met the inclusion criteria and were included in the systematic review (Figure 4).”
When discussing the evidences, I suggest to intensify the critics. Since there is not a clear strategy in the review of the literature, there is no real discussion regarding the limitations of current treatment modalities and diagnostic approaches.
Response: We thank the reviewer’s comment. We have added subsection 3.6 to discuss the current limitations on diagnosis and treatment. (pages 9-10, lines 326-351)
The statements read as”3.6. Current limitations on diagnosis and treatment
3.6.1. Diagnostic limitation
Due to its rarity, many clinicians and pathologists have limited experience, leading to frequent misdiagnosis. Histopathological confirmation requires neuroendocrine differentiation, but routine H&E staining may not be sufficient, necessitating immunohistochemical markers such as synaptophysin, chromogranin A, and CD56 [18]. However, inconsistent marker expression and a lack of standardized diagnostic criteria further complicate accurate identification. Molecular profiling, including TP53 and RB1 mutations, has shown a potential to improve diagnostic accuracy, but its clinical utility remains unstandardized [64]. Additionally, the absence of a universally accepted grading system makes it challenging to differentiate LCNEC from small-cell neuroendocrine carcinoma and other poorly differentiated tumors [5].
3.6.2. Therapeutic Limitations
Therapeutic options for LCNEC are limited, as there are no standardized treatment guidelines, and current approaches are often extrapolated from small-cell neuroendocrine carcinoma or pulmonary neuroendocrine tumors [65]. Radical hysterectomy is commonly performed, but surgery alone is insufficient due to early systemic dissemination [66]. Platinum-based chemotherapy (cisplatin/carboplatin with etoposide or irinotecan) remains the mainstay of treatment but is often ineffective in achieving long-term remission [59]. The role of adjuvant radiation therapy is unclear, with inconsistent survival benefits [59]. Emerging treatments, including immune checkpoint inhibitors and targeted therapies, have shown promise, but clinical data is lacking [57]. Given the high recurrence rate and poor prognosis, future research should focus on refining molecular diagnostics, conducting prospective clinical trials, and exploring novel therapeutic strategies to improve patient outcomes.”
correct some typos, enhance figures legend.
Response: We thank the reviewer’s comment. We have corrected typos and enhanced the figure legend.
The statements read as:”
Figure 1. Multimodal imaging of a pelvic lesion. (A) Transvaginal grayscale ultrasound image showing a well-defined solid mass measuring approximately 2.79 × 2.29 cm in the pelvic region. (B) Transvaginal color Doppler ultrasound demonstrated increased vascularization within the lesion, suggesting neovascularity. (C) Axial PET/CT fusion image highlights hypermetabolic activity within the lesion, indicating potential malignancy. (D) Coronal PET/CT fusion image further localizes the hypermetabolic pelvic mass about surrounding anatomical structures.
Figure 2. Gross pathology of the resected specimen. Macroscopic view of the surgically resected uterus, bilateral adnexa, and associated lymph nodes. The central specimen represents the uterus with an infiltrative tumor involving the endometrium, myometrium, and cervix. The surrounding excised tissues include the right (R) and left (L) adnexa and lymph nodes (LN), as well as an additional paraaortic (P) lymph node sample. The specimens are placed on a surgical mapping sheet for anatomical reference.
Figure 3. Histology and immunohistochemistry of the tumor. (A) Hematoxylin and eosin (H&E) staining at 40× magnification shows tumor architecture and cellular morphology. (B) High-power view (400× magnification) of the H&E-stained section, highlighting the tumor's cellular features. A nested and trabecular growth pattern with small, hyperchromatic tumor cells, scant cytoplasm, and frequent mitotic activity suggests an aggressive malignancy. (C) Immunohistochemical staining for synaptophysin at 100× magnification, demonstrating positive expression in tumor cells, indicative of neuroendocrine differentiation. (D) Immunohistochemical staining for CD56 at 100× magnification, showing strong membranous expression, further supporting the neuroendocrine nature of the tumor.)”
specify what kind “radical hysterectomy” they did according to the Querleu-Morrow classification (e.g., Type B, C, or D) and discuss around the approach (MIS vs open) considering that the cervical tumor measured 2.7 cm in diameter with hypervascularity given the evidence from the LACC trial.
Response: We thank the reviewer’s comment. We performed type C2 RH for the patient. We have added this information in the case part.
The statement reads as:”She underwent open type C2 radical hysterectomy (Querleu-Morrow classification), bilateral salpingo-oophorectomy…”(page 2, line 69)
We have also discussed the surgical approach for cervical cancer (MIS vs. open). (page 7, lines 216-226)
The statements read as:”The LACC trial demonstrated that minimally invasive surgery (MIS) for early-stage cervical cancer is associated with higher recurrence and mortality rates compared to open surgery [41,42]. This finding has led to a reevaluation of surgical approaches in cervical cancer management, with open radical hysterectomy now recommended as the standard of care for stage IA2-IB1 cervical cancer [41]. However, the adoption of this recommendation varies across Asian countries, with some still performing MIS in a significant proportion of cases [43]. Notably, MIS without uterine manipulator or with vaginal cuff closure showed similar recurrence rates to open surgery, suggesting potential modifications to improve MIS outcomes [43]. The LACC trial results have prompted further research and discussions among gynecologic oncologists worldwide, as evidenced by the Korean Society of Gynecologic Oncology survey [44].”
Reviewer 3 Report
Comments and Suggestions for Authors
The study is excellent, very interesting, always relevant. In this study of this rare tumor, we found all the necessary answers to all the questions when it comes to diagnosis and treatment and the follow-up of such a rare case.
We found answers to all these questions in this respected and top gynecologic work. The introduction, methodology, results, discussion and conclusions, literature, tables are superbly written and I recommend accepting this work and publishing it in a respected journal.
The only question for the case is the type of surgical approach, laparoscopy or laparotomy in the case mentioned, is not stated, but it is interesting.
I would like to discreetly shorten the discussion, that is, remove unnecessary repetitions of types of treatment.
Quality of English Language is excellent.
Author Response
The study is excellent, very interesting, always relevant. In this study of this rare tumor, we found all the necessary answers to all the questions when it comes to diagnosis and treatment and the follow-up of such a rare case.
We found answers to all these questions in this respected and top gynecologic work. The introduction, methodology, results, discussion and conclusions, literature, tables are superbly written and I recommend accepting this work and publishing it in a respected journal.
The only question for the case is the type of surgical approach, laparoscopy or laparotomy in the case mentioned, is not stated, but it is interesting.
Response: We thank the reviewer’s comment. We have provided the surgery method in this case (open). (page 2, line 69)
The statements read as:”She underwent open type C2 radical hysterectomy (Querleu-Morrow classification), bilateral salpingo-oophorectomy….”
I would like to discreetly shorten the discussion, that is, remove unnecessary repetitions of types of treatment.
Response: We thank the reviewer’s comment. We have removed the repetition of types of treatment. (Section 3.4, pages 7-9, lines 203-284)